# Spatial epidemiology of gestational age and birth weight in Switzerland: census-based linkage study

Veronika Skrivankova,[1] Marcel Zwahlen [ID],[1] Mark Adams,[2] Nicola Low [ID],[1] Claudia Kuehni,[1,3] Matthias Egger [ID][1]

[1]Institute of Social and Preventive Medicine, University of Bern, Bern, Switzerland
[2]Department of Neonatology, University of Zurich, Zurich, Switzerland
[3]Department of Pediatrics, Inselspital, Bern University Hospital, University of Bern, Bern, Switzerland

**Correspondence to**
Dr Matthias Egger;
matthias.egger@ispm.unibe.ch

## ABSTRACT

**Background** Gestational age and birth weight are strong predictors of infant morbidity and mortality. Understanding spatial variation can inform policies to reduce health inequalities. We examined small-area variation in gestational age and birth weight in Switzerland.

**Methods** All singleton live births recorded in the Swiss Live Birth Register 2011 to 2014 were eligible. We deterministically linked the Live Birth Register with census and survey data to create data sets including neonatal and pregnancy-related variables, parental characteristics and geographical variables. We produced maps of 705 areas and fitted linear mixed-effect models to assess to what extent spatial variation was explained by these variables.

**Results** We analysed all 315 177 eligible live births. Area-level averages of gestational age varied between 272 and 279 days, and between 3138 and 3467 g for birth weight. The fully adjusted models explained 31% and 87% of spatial variation of gestational age and birth weight, respectively. Language region accounted for most of the explained variation (23% in gestational age and 62% in birth weight), with shorter gestational age (−0.6 days and −0.9 days) and lower birth weight (−1.1% and −1.8%) in French-speaking and Italian-speaking areas, respectively, compared with German-speaking areas. Other variables explaining variation were, for gestational age, the level of urbanisation (10%) and parental nationality (3%). For birth weight, they were gestational age (27%), parental nationality (27%), civil status (10%) and altitude (10%). In a random sample of 81 968 live births with data on parental education, levels of education were only weakly associated with gestational age (−0.9 days for compulsory vs tertiary maternal education) or birth weight (−0.7% for compulsory vs tertiary maternal education).

**Conclusions** In Switzerland, small area variation in birth weight is largely explained, and variation in gestational age partially explained, by geocultural, sociodemographic and pregnancy factors.

## INTRODUCTION

Gestational age and birth weight are important indicators of prenatal development and predictors of infant morbidity, mortality and long-term health.[1–4] An understanding of geographic differences and their determinants can help to develop policies that reduce health inequalities across population

### Strengths and limitations of this study

► This study was based on a large sample with national coverage and routinely collected data on neonatal and pregnancy-related predictors of gestational age and birth weight.
► Precise location data allowed for detailed geographical maps of spatial distribution and assessment of spatial variation in the two birth outcomes.
► No data were available on the mode of delivery, health-related behaviours such as maternal smoking or gestational diabetes.
► Parental nationality served as crude proxy for parental height and weight, and language region as a proxy for a range of cultural, social and behavioural factors.

groups and regions.[1–4] Many genetic, physiological, pregnancy-related, socioeconomic, lifestyle and environmental factors have been reported to influence gestational age and birth weight.[5–8] Some of these factors tend to cluster in space and regional differences in health outcomes may hence be partially explained by the spatial distribution of their predictors. Importantly, both individual-level factors and the social and environmental characteristics of communities and neighbourhoods may contribute to regional differences.[9 10]

Variation across small areas in pregnancy outcomes have not been studied widely. In Scotland, small-area crime rates were associated with lower birth weight and with the risk of both small for gestational age babies and preterm birth.[11] A study at county level in Georgia and South Carolina in the USA showed that the proportion of African Americans was associated with low birth weight, whereas higher income was associated with higher birth weight.[12] Similarly, neighbourhood racial composition contributed to variation in low birth weight in New York State.[13] Other small-area analyses have examined associations between birth outcomes and air

pollution.[14 15] To our knowledge, few small-area analyses have considered gestational age.

In Switzerland, studies of pregnancy outcomes have focused on specific groups such as migrants or HIV-infected women,[16 17] but have not examined geographic variations. The Federal Office of Statistics publishes routine statistics from the Live Birth Register, which does not include geographic information.[18] The objectives of this study were to conduct a nationwide analysis of spatial variation in gestational age and birth weight, and to assess how much small-area variation was explained by available data about neonatal and pregnancy-related variables, parental characteristics and geographical variables.

## METHODS

### Data sources

We used deterministic methods to link three data sources using encrypted national identification numbers: the Live Birth Register, the Swiss National Cohort and the Structural Surveys. Registration of live births is compulsory by law in Switzerland and coverage is near 100%. The Swiss National Cohort (SNC) is a long-term, national study of mortality in Switzerland,[19 20] linking census and mortality records. The 1990 and 2000 censuses were the last house-to-house censuses with coverage of the entire Swiss population. From 2010 onwards, the national census was replaced by a national population register and annual postal survey of the resident population, known as Structural Surveys.[21] Each Structural Survey includes a random sample of around 300 000 people aged 15 years or older; for example, in 2010, it included 317 221 persons.[21] The reference is the entire Swiss resident population and the reference day December 31.

### Variables and definitions

We defined three sets of variables. The first set, neonatal and pregnancy-related variables come from the Live Birth Register; date of birth, birth weight, gestational age, sex and birth rank. Birth weight is measured after initial mother–child bonding, usually by the midwife using a calibrated hospital scale. Gestational age is based on the last menstrual period, with or without additional information from ultrasound scans. Birth rank was determined from the list of all live births by the same mother recorded in the Live Birth Register, and is hence restricted to the births that occurred in Switzerland. It was classified as 1, 2, 3 and ≥4 live births, including the current birth. The second set includes parental variables. The Structural Surveys provide information about the highest level of completed maternal and paternal education, classified as 'tertiary', 'secondary' or 'compulsory or less'. The Swiss National Cohort provides data about parental nationality categorised as 'Swiss', 'Southern Europe', 'Western Europe', 'Northern Europe', 'Eastern Europe', 'Other' (non-European) or missing (online supplementary table S1 gives the full list of countries). The third set, geographical variables comes from the Swiss National Cohort. Each

live birth was assigned an altitude and 1 of 705 statistical areas,[22] based on the geocode of place of residence of the mother at the time of birth. Language regions are 'German', 'French' and 'Italian', and the level of urbanisation was defined using standard definitions of 'urban', 'peri-urban' and 'rural'.

### Study populations and outcomes

All singleton live births recorded in the Live Birth Register from 1 January 2011 to 31 December 2014 were eligible. Gestational age at birth and birth weight were the outcomes of interest. For each outcome, two data sets were analysed: the first, larger data set consisted of all eligible births with complete data on gestational age, birth weight and nationality of the mother. The second was the complete case population containing eligible live births with available data on all variables, including parental education. The second data set hence included only newborns whose parents were included in the random sample of one of the Structural Surveys 2010–2014. We also excluded mothers who delivered at age less than 20 years, because education is incomplete at that age.

### Statistical and spatial analyses

We fitted linear mixed-effect models (LMEM) to examine the associations between the two outcomes and the neonatal and pregnancy, parental and environmental factors. In the model for birth weight, we log-transformed the outcome and used a cubic spline function with three knots at weeks 25, 30 and 35 to capture the relationship between gestational age and log birth weight. Log transforming the birth weight results in a multiplicative model. Except for gestational age, maternal age and altitude, all predictors were modelled categorically. Maternal age was modelled by a piece-wise linear function, with age group 20 to 30 years as the reference group and separate linear trends for age groups 30–40 years, over 40 years and less than 20 years. Altitude was centred at 500 m and modelled linearly. The random effects in the mixed-effect model captured area-level differences between observed and expected mean outcome, based on the 705 statistical areas.[22] In the main analysis, we fitted four models to the complete-case data set: Model 0 contained no explanatory variables. Model 1 included birth and pregnancy-related variables: sex, birth rank and gestational age (for the analysis of birth weight). Model 2 additionally included age of the mother, parental education and nationality. Model 3 additionally included geographical variables: altitude, degree of urbanisation and language region.

We displayed mean gestational age and birth weight at area level on maps and assessed to what extent spatial variation was accounted for by the explanatory variables. Values were categorised into seven intervals symmetric around the mean and color-coded. Spatial autocorrelation of the gestational age and birth weight across regions was tested by global and local Moran's I tests.[23] The global Moran test summarises overall spatial autocorrelation and the local test identifies areas that are correlated with

**Table 1** Characteristics of complete case and eligible study populations

| | Eligible population | | | Complete case population | | |
|---|---|---|---|---|---|---|
| | | Gest. age (days) | Birth weight (g) | | Gestational age (days) | Birth weight (g) |
| | No. (%) | Mean (SD) | Mean (SD) | No. (%) | Mean (SD) | Mean (SD) |
| **Total** | 315 177 (100%) | 276 (12) | 3328 (515) | 81 968 (100%) | 276 (12) | 3339 (501) |
| **Birth weight (g)** | | | | | | |
| <1500 | 2141 (0.7%) | 196 (27) | 966 (354) | 445 (0.5%) | 198 (28) | 983 (491) |
| 1500–1999 | 2413 (0.8%) | 238 (15) | 1800 (142) | 612 (0.7%) | 239 (15) | 1803 (528) |
| 2000–2499 | 10 036 (3.2%) | 258 (14) | 2312 (134) | 2484 (3%) | 258 (13) | 2314 (477) |
| ≥2500 | 300 586 (95.4%) | 277 (9) | 3391 (423) | 78 426 (95.7%) | 277 (9) | 3396 (502) |
| **Gestational age (weeks)** | | | | | | |
| <$32^0$ | 2333 (0.7%) | 195 (23) | 1108 (527) | 487 (0.6%) | 196 (23) | 1134 (491) |
| $32^0$–$34^6$ | 3950 (1.3%) | 237 (6) | 2144 (424) | 961 (1.2%) | 237 (6) | 2136 (470) |
| $35^0$–$36^6$ | 10 907 (3.5%) | 253 (4) | 2686 (431) | 2760 (3.4%) | 253 (4) | 2692 (489) |
| ≥$37^0$ | 297 987 (94.5%) | 278 (8) | 3385 (440) | 77 760 (94.9%) | 278 (8) | 3390 (502) |
| **Sex** | | | | | | |
| Female | 152 757 (48.5%) | 276 (12) | 3260 (494) | 39 823 (48.6%) | 276 (11) | 3267 (502) |
| Male | 162 420 (51.5%) | 275 (13) | 3392 (525) | 42 145 (51.4%) | 276 (12) | 3406 (501) |
| **Birth rank** | | | | | | |
| 1 | 155 739 (49.4%) | 276 (13) | 3262 (519) | 37 763 (46.1%) | 276 (13) | 3267 (498) |
| 2 | 115 440 (36.6%) | 275 (11) | 3382 (497) | 32 315 (39.4%) | 276 (11) | 3386 (504) |
| 3 | 34 364 (10.9%) | 275 (11) | 3418 (509) | 9360 (11.4%) | 275 (11) | 3430 (508) |
| ≥4 | 9634 (3.1%) | 275 (12) | 3438 (537) | 2530 (3.1%) | 275 (11) | 3459 (498) |
| **Civil status** | | | | | | |
| Married | 250 055 (79.3%) | 276 (12) | 3345 (508) | 69 465 (84.7%) | 276 (12) | 3349 (501) |
| Not married | 65 122 (20.7%) | 276 (14) | 3262 (536) | 12 503 (15.3%) | 276 (13) | 3283 (503) |
| **Maternal age (years)** | | | | | | |
| Mean (SD) | *31.7 (5.0)* | | | *32.2 (4.7)* | | |
| <20 | 2679 (0.8%) | 275 (16) | 3224 (554) | 0 (0%) | – | – |
| ≥20–25 | 28 615 (9.1%) | 277 (12) | 3317 (511) | 5417 (6.6%) | 277 (12) | 3337 (491) |
| ≥25–30 | 82 620 (26.2%) | 276 (12) | 3330 (506) | 20 771 (25.3%) | 276 (12) | 3337 (500) |
| ≥30–35 | 118 303 (37.5%) | 276 (12) | 3335 (510) | 32 771 (40%) | 276 (12) | 3341 (505) |
| ≥35–40 | 67 914 (21.5%) | 275 (12) | 3333 (523) | 19 052 (23.2%) | 275 (11) | 3345 (497) |
| ≥40 | 15 046 (4.8%) | 273 (14) | 3286 (555) | 3957 (4.8%) | 273 (14) | 3295 (512) |
| **Nationality mother** | | | | | | |
| Switzerland | 194 570 (61.7%) | 276 (12) | 3322 (511) | 55 591 (67.8%) | 276 (12) | 3331 (502) |
| Southern Europe | 23 585 (7.5%) | 275 (12) | 3251 (494) | 5761 (7%) | 276 (11) | 3261 (502) |
| Western Europe | 26 005 (8.3%) | 276 (12) | 3348 (516) | 6495 (7.9%) | 276 (12) | 3359 (508) |
| Northern Europe | 3695 (1.2%) | 276 (13) | 3418 (510) | 850 (1%) | 276 (13) | 3414 (508) |
| Eastern Europe | 38 762 (12.3%) | 276 (13) | 3397 (523) | 8035 (9.8%) | 276 (12) | 3422 (499) |
| Other | 28 560 (9.1%) | 275 (14) | 3313 (535) | 5236 (6.4%) | 275 (13) | 3332 (492) |
| **Nationality father** | | | | | | |
| Switzerland | 191 589 (60.8%) | 276 (12) | 3329 (506) | 55 432 (67.6%) | 276 (12) | 3336 (502) |
| Southern Europe | 31 466 (10%) | 275 (12) | 3256 (493) | 7970 (9.7%) | 275 (11) | 3262 (504) |
| Western Europe | 26 954 (8.6%) | 276 (12) | 3353 (518) | 6661 (8.1%) | 276 (12) | 3367 (514) |
| Northern Europe | 3911 (1.2%) | 276 (12) | 3406 (510) | 887 (1.1%) | 276 (13) | 3393 (499) |
| Eastern Europe | 35 387 (11.2%) | 276 (13) | 3397 (528) | 7229 (8.8%) | 276 (12) | 3418 (489) |

Continued

**Table 1** Continued

| | Eligible population | | | Complete case population | | |
|---|---|---|---|---|---|---|
| | No. (%) | Gest. age (days) Mean (SD) | Birth weight (g) Mean (SD) | No. (%) | Gestational age (days) Mean (SD) | Birth weight (g) Mean (SD) |
| Other | 21 077 (6.7%) | 276 (13) | 3307 (531) | 3789 (4.6%) | 276 (12) | 3319 (497) |
| Missing | 4793 (1.5%) | 272 (23) | 3148 (693) | – | – | – |
| **Education mother** | | | | | | |
| Tertiary | 42 088 (13.4%) | 276 (12) | 3344 (500) | 33 505 (40.9%) | 276 (12) | 3347 (500) |
| Secondary | 48 878 (15.5%) | 276 (12) | 3328 (509) | 38 382 (46.8%) | 276 (12) | 3331 (502) |
| Compulsory | 14 642 (4.6%) | 275 (13) | 3329 (534) | 10 081 (12.3%) | 275 (13) | 3336 (503) |
| Unknown (age <20 years) | 2679 (0.8%) | 275 (16) | 3224 (554) | 0 (0%) | – | – |
| Missing | 206 890 (65.6%) | 276 (12) | 3326 (517) | – | – | – |
| **Education father** | | | | | | |
| Tertiary | 49 848 (15.8%) | 276 (12) | 3348 (497) | 40 345 (49.2%) | 276 (12) | 3350 (500) |
| Secondary | 41 301 (13.1%) | 276 (12) | 3323 (511) | 32 118 (39.2%) | 276 (12) | 3327 (504) |
| Compulsory | 13 731 (4.4%) | 276 (12) | 3323 (514) | 9505 (11.6%) | 276 (12) | 3330 (500) |
| Missing | 210 297 (66.7%) | 276 (13) | 3325 (519) | – | – | – |
| **Altitude (m)** | | | | | | |
| Mean (SD) | *515 (189)* | | | *511 (180)* | | |
| **Urbanisation** | | | | | | |
| Urban | 96 643 (30.7%) | 276 (13) | 3326 (517) | 22 770 (27.8%) | 276 (12) | 3334 (502) |
| Peri-urban | 138 826 (44%) | 275 (12) | 3329 (514) | 36 629 (44.7%) | 276 (12) | 3339 (502) |
| Rural | 79 708 (25.3%) | 276 (12) | 3329 (512) | 22 569 (27.5%) | 276 (12) | 3343 (500) |
| **Language region** | | | | | | |
| German | 223 586 (70.9%) | 276 (12) | 3348 (515) | 54 106 (66%) | 276 (12) | 3362 (502) |
| French | 80 068 (25.4%) | 275 (12) | 3283 (512) | 23 579 (28.8%) | 275 (12) | 3296 (501) |
| Italian | 11 523 (3.7%) | 275 (12) | 3252 (494) | 4283 (5.2%) | 275 (11) | 3268 (500) |
| **Socioeconomic position** | | | | | | |
| First quintile | 63 230 (20.1%) | 276 (12) | 3318 (522) | 15 752 (19.2%) | 276 (12) | 3331 (501) |
| Second quintile | 63 199 (20.1%) | 276 (12) | 3324 (519) | 16 034 (19.6%) | 276 (12) | 3334 (505) |
| Tthird quintile | 63 156 (20%) | 276 (12) | 3329 (516) | 16 555 (20.2%) | 276 (12) | 3337 (500) |
| Fourth quintile | 62 970 (20%) | 276 (12) | 3335 (509) | 16 933 (20.7%) | 276 (12) | 3344 (500) |
| Fifth quintile | 62 622 (19.9%) | 276 (12) | 3335 (507) | 16 694 (20.4%) | 276 (12) | 3346 (502) |

neighbouring areas. In the presence of spatial autocorrelation, model estimates are at risk of bias if the autocorrelation is not taken into account.

We performed three sensitivity analyses. First, we accounted for spatial autocorrelation using the Besag-York-Mollier (BYM) model[24] using uninformative gamma-distributed (1, 0.005) priors. The calculations were carried out using the Integrated Nested Laplace Approximation (INLA) approach.[25] Similar results from models with and without the spatial component indicate low bias. Second, we repeated analyses of birth weight without adjusting for gestational age. Third, we repeated analyses of birth weight and gestational age, additionally adjusting for neighbourhood socioeconomic position (SEP), using an updated version of the Swiss SEP index, which is based on levels of rent, education and occupation of heads of

households and crowding.[26] The updated version of the index is based on data from Structural Surveys 2010–2014, and includes information on income of households in the neighbourhood. We used quintiles of the index in the analysis, with higher quintiles indicating higher SEP.

All analyses and maps were done in R V.3.3.2[27] using packages lme4, maptools, sp, spdep, rgdal, INLA, GISTools, rgeos, raster and ggplot2.

## Patient and public involvement

This analysis was based on routine registry data and no patients were involved in developing the research question, outcome measures and overall design of the study. Due to the anonymous nature of the data, we were unable to disseminate the results of the research directly to study participants.

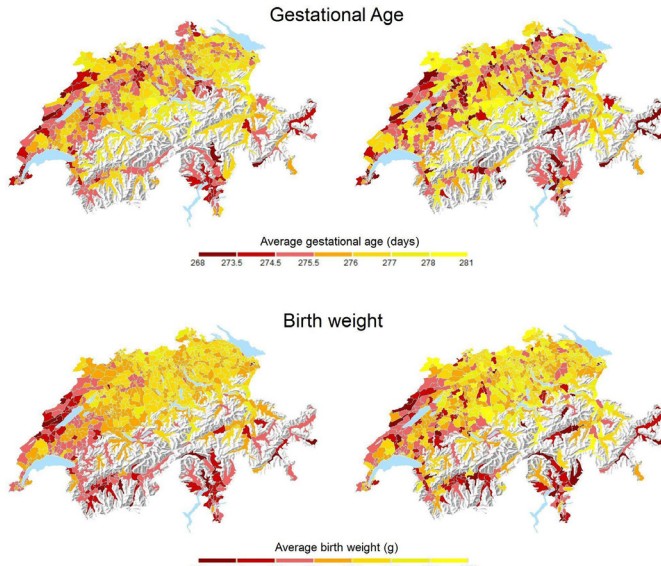

**Figure 1** Maps of average gestational age (upper two panels) and birth weight (lower two panels) observed across 705 Swiss areas. Left: all eligible live births (n=3 15 177), right: complete case population (n=81 968). Note that 277 days correspond to 39$^{4/7}$ weeks. The orientation of the maps is standard, with North being up.

## RESULTS
### Characteristics of study populations

A total of 328 349 live births were recorded in Switzerland between 1 January 2011 and 31 December 2014. We excluded non-singleton live births (n=11 835) and those with missing gestational age, birth weight or maternal nationality. The eligible study population therefore included 315 177 singleton live births. The complete case population consisted of 81 968 singleton live births with values available for all predictors including parental education, for which complete data were only available in the Structural Surveys (online supplementary figure S1).

Table 1 shows the distributions of predictors and outcomes in the two study populations. Data about the nationality of fathers were missing for 1.5% of eligible live births. In almost all of these cases, information about the father was missing completely, indicating that the father is unknown to the authorities. Apart from missing data, the distributions of most variables were similar between the two nested data sets. The proportion of Swiss mothers and fathers was higher in the complete case population than in the eligible population. Birth at full term is defined as between 39 and 41 weeks of gestation (273 to 287 days). The mean gestational age in the eligible population was 276 days (SD 12) and the mean birth weight 3328 g (SD 515). The corresponding figures in the complete case population were 276 days (SD 12) and 3339 g (SD 501). Maps of gestational age and birth weight

Figure 1 presents maps of Switzerland with crude average gestational age and birth weight across the 705 areas. For both outcomes, the maps are broadly similar between the eligible and complete case populations. For gestational age, area-level averages for the eligible

population vary between 272 and 279 days. For the complete case population, variation was greater, from 268 to 281 days, as expected for a smaller sample. The map shows shorter gestation in the Western, North Western region and Southern (Canton of Ticino) regions of Switzerland, with a patchy pattern in the densely populated areas between the Alps (across the centre) and Jura mountain ranges (to the North West). For birth weight, area-level averages vary between 3138 and 3467 g for the eligible population and between 3080 and 3648 g for the complete case population. The maps for birth weight show lower birth weights in the Western and Southern regions of the country. The French and Italian-speaking regions are in the West and South of Switzerland, with the remainder being German-speaking.

### Multivariable analyses

Table 2 shows associations of area-level mean gestational age at birth and mean birth weight with pregnancy, parental and environmental factors from the fully adjusted linear mixed-effects models (model 3). For gestational age, the largest differences were observed across maternal age at birth. Compared with maternal age 20–30 years, gestational age was considerably shorter in teenage mothers, and in mothers aged over 40 years. For example, in mothers aged 15 years, pregnancies were about 4 days shorter, and after age of 40 years, they were about 3 days shorter for each 5 year increase in maternal age. Compared with Swiss fathers, pregnancies were about 4 days shorter if the nationality of the father was missing. Smaller differences in gestational age were observed across categories of sex, birth rank, nationality of the mother, urbanisation and between language regions (table 2). In the complete case population, lower levels of education were associated with shorter pregnancies. Gestational age at birth was not associated with altitude.

Online supplementary figure 2 shows the relationship between gestational age and birth weight separately for male and female newborns. Male newborns were about 5% heavier than female newborns and birth weight increased with birth order (table 2). In contrast to gestational age, mother's age was not associated with birth weight. Babies born to mothers or fathers from Northern or Eastern Europe were slightly heavier than babies born to Swiss mothers; birth weights were lowest for babies of fathers with missing nationality. Birth weight slightly decreased with increasing parental educational attainment. Babies born in the French and Italian-speaking regions were lighter than babies born in the German-speaking Switzerland. Finally, birth weight decreased with increasing altitude of residence.

### Proportion of spatial variation explained

The fully adjusted model (model 3) for gestational age explained 31% and 39% of the spatial variation across the 705 areas for eligible and complete case populations, respectively. The corresponding figures for birth weight were 87% and 88%. When assessing each factor separately

**Table 2** Associations of mean gestational age at birth and mean birth weight with pregnancy, parental and environmental factors from adjusted linear mixed-effects model (model 3)

| | Gestational age (days) absolute differences (95% CI) | | Birth weight (g) * relative differences (95% CI) | |
| --- | --- | --- | --- | --- |
| | Eligible population | Complete case population | Eligible population | Complete case population |
| **Intercept** | 277.3 (277.2 to 277.5) | 277.9 (277.7 to 278.2) | 3278 (3218 to 3339)† | 3298 (3180 to 3420)† |
| **Sex** | | | | |
| Female | 0 | 0 | 1 | 1 |
| Male | −0.56 (-0.65 to -0.48) | −0.63 (-0.79 to -0.47) | 1.045 (1.044 to 1.046) | 1.048 (1.046, 1.049) |
| **Birth rank** | | | | |
| 1‡ | 0 | 0 | 1 | 1 |
| 2 | −0.39 (-0.49 to -0.29) | −0.34 (-0.52 to -0.16) | 1.038 (1.037 to 1.039) | 1.039 (1.037, 1.041) |
| 3 | −0.37 (-0.52 to -0.22) | −0.16 (-0.44 to 0.11) | 1.050 (1.048 to 1.051) | 1.054 (1.051, 1.057) |
| ≥4 | −0.24 (-0.50 to 0.02) | 0.24 (-0.25 to 0.72) | 1.058 (1.056 to 1.061) | 1.065 (1.059, 1.070) |
| **Age mother (years)§** | | | | |
| <20 (per 5 years decr.) | −4.10 (-5.59 to -2.61) | – | 1.002 (0.987 to 1.017) | – |
| ≥20–30‡ | 0 | 0 | 1 | 1 |
| ≥30–40 (per 5 years) | −0.99 (-1.06 to -0.91) | −0.93 (-1.07 to -0.78) | 1.000 (1.000 to 1.001) | 0.998 (0.997, 1.000) |
| ≥40 (per 5 years) | −2.93 (-3.36 to -2.50) | −3.46 (-4.29 to -2.63) | 0.998 (0.994 to 1.003) | 0.998 (0.990, 1.006) |
| **Civil status¶** | | | | |
| Married | 0 | 0 | 1 | 1 |
| Not married | −0.01 (-0.13 to 0.10) | 0.15 (-0.08 to 0.38) | 0.990 (0.989 to 0.991) | 0.993 (0.99, 0.995) |
| **Nationality mother** | | | | |
| Switzerland‡ | 0 | 0 | 1 | 1 |
| S Europe | 0.20 (-0.01 to 0.40) | 0.39 (00 to 0.78) | 0.994 (0.992 to 0.996) | 0.995 (0.991, 0.999) |
| W Europe | 0.20 (0.02 to 0.38) | −0.08 (-0.43 to 0.26) | 1.008 (1.006 to 1.010) | 1.007 (1.004, 1.011) |
| N Europe | 0.37 (-0.07 to 0.81) | 0.30 (-0.57 to 1.17) | 1.025 (1.020 to 1.029) | 1.022 (1.013, 1.031) |
| E Europe | 0.21 (0.04 to 0.38) | 0.33 (-0.01 to 0.68) | 1.013 (1.011 to 1.014) | 1.017 (1.014, 1.021) |
| Other | −0.32 (-0.49 to -0.14) | −0.67 (-1.05 to -0.30) | 1.007 (1.005 to 1.008) | 1.012 (1.008, 1.016) |
| **Nationality father** | | | | |
| Switzerland‡ | 0 | 0 | 1 | 1 |
| S Europe | −0.46 (-0.64 to -0.28) | −0.28 (-0.62 to 0.06) | 0.991 (0.990 to 0.993) | 0.993 (0.989, 0.996) |
| W Europe | 0.07 (-0.11 to 0.25) | 0.30 (-0.04 to 0.63) | 1.008 (1.006 to 1.009) | 1.006 (1.003, 1.010) |
| N Europe | 0.51 (0.08 to 0.94) | −0.24 (-1.09 to 0.62) | 1.013 (1.009 to 1.017) | 1.011 (1.003, 1.020) |
| E Europe | −0.46 (-0.64 to -0.28) | −0.01 (-0.38 to 0.36) | 1.009 (1.007 to 1.010) | 1.011 (1.008, 1.015) |
| Other | −0.02 (-0.22 to 0.18) | 0.48 (0.05 to 0.90) | 0.992 (0.991 to 0.994) | 0.992 (0.987, 0.996) |
| Missing | −3.87 (-4.24 to -3.50) | – | 0.989 (0.985 to 0.992) | – |
| **Education mother** | | | | |
| Tertiary‡ | | 0 | | 1 |
| Secondary | | −0.55 (-0.74 to -0.36) | | 0.996 (0.995, 0.998) |
| Compulsory | | −0.90 (-1.22 to -0.58) | | 0.993 (0.990, 0.996) |
| **Education father** | | | | |
| Tertiary‡ | | 0 | | 1 |
| Secondary | | −0.16 (-0.35 to 0.03) | | 0.996 (0.994, 0.998) |
| Compulsory | | −0.25 (-0.58 to 0.07) | | 0.997 (0.994, 1.000) |
| **Altitude (m)** | | | | |
| 500‡ | 0 | 0 | 1 | 1 |
| per 500 m increase | 0.07 (-0.09 to 0.23) | 0.03 (-0.24 to 0.30) | 0.989 (0.988 to 0.991) | 0.989 (0.987, 0.992) |
| Urbanisation | | | | |

Continued

**Table 2** Continued

| | Gestational age (days) absolute differences (95% CI) | | Birth weight (g) * relative differences (95% CI) | |
| --- | --- | --- | --- | --- |
| | **Eligible population** | **Complete case population** | **Eligible population** | **Complete case population** |
| Urban‡ | 0 | 0 | 1 | 1 |
| Peri-urban | −0.43 (−0.57 to -0.28) | −0.59 (−0.82 to -0.36) | 1.001 (1.000 to 1.002) | 1.003 (1.000, 1.005) |
| Rural | −0.15 (−0.32 to 0.02) | −0.29 (−0.55 to -0.02) | 1.000 (0.998 to 1.001) | 1.003 (1.001, 1.006) |
| **Language region** | | | | |
| German‡ | 0 | 0 | 1 | 1 |
| French | −0.62 (−0.77 to -0.47) | −0.66 (−0.88 to -0.44) | 0.989 (0.987 to 0.990) | 0.988 (0.985, 0.990) |
| Italian | −0.94 (−1.26 to -0.63) | −1.11 (−1.55 to -0.68) | 0.982 (0.980 to 0.985) | 0.983 (0.979, 0.987) |
| **Percent of spatial variance explained**\*\* | 31% | 39% | 87% | 88% |

*Birth weight was modelled on a log scale, which results in multiplicative effects. The model for birth weight was additionally adjusted for gestational age by a cubic spline function with knots at weeks 25, 30 and 35.
†In the model for BW, the intercept corresponds to an estimated mean birth weight (g) for a singleton girl born at gestational age 40 weeks as the first child (rank 1) in a German-speaking, urban region of elevation 500m, whose mother is 20-30 years old at birth and married, and both parents have Swiss nationality and tertiary education.
‡Reference category.
§Age modelled by a piece-wise linear function: constant at reference range ≥20-30, and separate slopes for age <20, ≥30-40, and ≥40.
¶Married or in registered partnership / Not married: Single, widow, divorced or in dissolved partnership.
**Percentage of regional variance explained by model predictors, i.e. percent reduction in variance of random effects ($\sigma^2$) when compared to model with no predictors (model 0).

(table 3), language region alone explained most of the spatial variation for both outcomes. For gestational age, level of urbanisation of the mother's place of residence also explained a considerable part of the variation. Factors that contributed to explaining the spatial variation in birth weight were gestational age, parental nationalities, altitude at the mother's place of residence and birth order. figure 2 illustrates the reduction in the spatial variation of gestational age and birth weight with maps, when moving from model 0 (0% reduction) to models 1, 2 and 3, based on the complete case population.

### Spatial autocorrelation and sensitivity analyses

For gestational age, the global Moran's I statistic, based on the complete case data set and model 0, was I=0.19, with $p<10^{-14}$. After adjusting for all the predictors in model 3, there was still some residual autocorrelation (I=0.10, p=0.0004). For birth weight, the corresponding Moran's I statistic was I=0.28, with $p<10^{-15}$. After adjusting for all predictors in model 3 there was little residual autocorrelation (I=0.04, p=0.051). Online supplementary table S2 compares the results from model 3 accounting and not accounting for spatial autocorrelation. The results are similar and the potential bias from residual spatial autocorrelation is therefore unlikely to be a major issue. Repeating analyses of birth weight without adjusting for gestational age produced generally similar coefficients (online supplementary table S3). Associations with maternal age, maternal education and language regions were slightly stronger in model 3 without adjustment for gestational age, possibly because some of their effect was mediated by gestational age. Model 3 without gestational

age explained 77% and 76% of the spatial variation in the eligible and complete case population, respectively. The index of neighbourhood SEP was only weakly associated with the two outcomes (online supplementary table S4), and adjusting for it only slightly increased the amount of spatial variation explained.

### DISCUSSION

Our study assessed factors associated with gestational age and birth weight in Switzerland and their contribution to spatial variation, based on routinely collected data. Gestational age at birth was strongly associated with maternal age, missing information on the father and language region. Birth weight was associated with sex, birth rank, missing information on the father, parental education, altitude and language region. The variables included in the fully adjusted model explained more than 80% of the regional variation in birth weight and about 30% of the regional variation in gestational age. Strengths of this study include a large sample with national coverage of the Swiss resident population, as well as the availability of data on several relevant predictors, either on all births or on a large random sample of eligible births. Precise spatial data and spatial statistics allowed us to assess the proportion of area-level variation explained, spatial autocorrelation and gauge the likelihood of bias due to residual autocorrelation.

This study found important spatial variation in both gestational age and birth weight in Switzerland. Language region in Switzerland was the single factor that explained

**Table 3** Percentage of spatial variation explained by each individual variable and explained in addition after adjusting for all other variables

| Spatial variation explained | Gestational age | | Birth weight | |
|---|---|---|---|---|
| | Eligible population | Complete case population | Eligible population | Complete case population |
| **By single variables** | | | | |
| Pregnancy factor | | | | |
| Gestational age | – | – | 27% | 34% |
| Sex | 0% | 0% | 1% | 2% |
| Birth rank | 0% | 1% | 4% | 0% |
| Parental factors | | | | |
| Maternal age | 0% | 1% | 1% | 1% |
| Civil status | 0% | 0% | 10% | 5% |
| Nationality mother | 1% | 3% | 17% | 17% |
| Nationality father | 3% | 4% | 25% | 20% |
| Nationality parents* | 3% | 5% | 27% | 23% |
| Education mother | – | 1% | – | 0% |
| Education father | – | 1% | – | 1% |
| Education parents* | – | 1% | – | 1% |
| Regional factors | | | | |
| Altitude | 0% | 0% | 10% | 6% |
| Urbanisation | 10% | 12% | 0% | 0% |
| Language region | 23% | 25% | 62% | 63% |
| **In addition to all other variables** | | | | |
| Pregnancy factors | | | | |
| Gestational age | – | – | 12% | 12% |
| Sex | 0% | 0% | 0% | 1% |
| Birth rank | 1% | 0% | 3% | 1% |
| Parental factors | | | | |
| Maternal age | 0% | 1% | 0% | 0% |
| Civil status | 0% | 0% | 0% | 0% |
| Nationality mother | 0% | 0% | 1% | 2% |
| Nationality father | 1.5% | 0% | 1% | 0% |
| Nationality parents* | 2.5% | 0% | 3% | 4% |
| Education mother | – | 2% | – | 0% |
| Education father | – | 0% | – | 0% |
| Education parents | – | 2% | – | 1% |
| Regional factors | | | | |
| Altitude | 0% | 0% | 9% | 4% |
| Urbanisation | 9% | 10% | 0% | 1% |
| Language region | 17% | 21% | 22% | 24% |
| **Model 3 (full)** | **31%** | **39%** | **87%** | **88%** |

–Data not available
*Nationality or educational attainment of both mother and father were entered into the model.

the greatest proportion of spatial variation in gestational age and birth weight. In the French-speaking and Italian-speaking regions, gestational age was shorter and birth weight lower than in the German speaking part. Language region is a proxy for a wide range of cultural, social and behavioural factors, including diet, smoking and alcohol consumption[28] of parents, as well as their ancestry. In

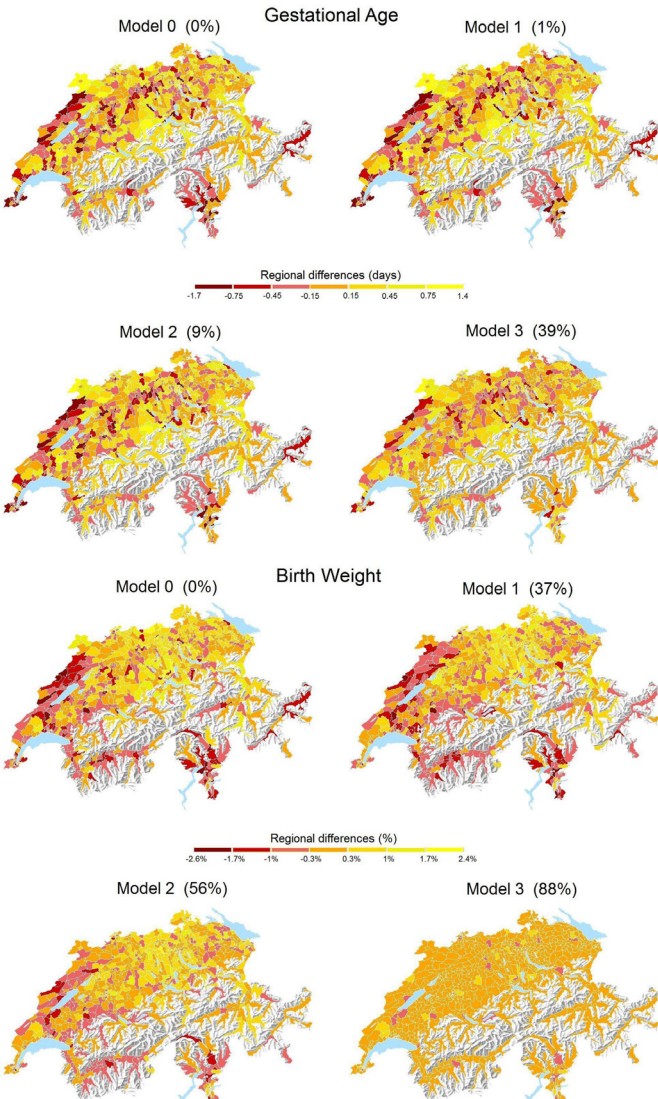

**Figure 2** Maps of gestational age and birth weight from crude model (model 0) and multivariable linear mixed-effect models (models 1–3) with percent reduction in the regional variation, represented by random effects. Analyses based on complete case population (n=81 968). The orientation of the maps is standard, with North being up.

this context, it is noteworthy that neighbourhood SEP explained only a small proportion of the spatial variation.

Other factors that could not be measured directly, such as healthcare provision, might have accounted for some of the unexplained variation. In particular, data at the individual or small area level on the mode of delivery (vaginal or by Caesarean section, induced or spontaneous) were not available. The proportion of live births with Caesarian section as the mode of delivery varies across regions in Switzerland, and it is reasonable to expect that it would explain some of the remaining variation, both in gestational age and birth weight. Specifically, we would expect regions with higher proportions of Caesarian section to have lower mean gestational age (and consequently birth weight). However, the regional rates of Caesarian section published by the Federal Office of Statistics do not match

this expectation,[29] with urban areas showing some of the highest Caesarian section rates but also high mean gestational age and birth weight. In fact, geographical patterns of Caesarean section seem to be largely driven by urban-rural differences. Differences in section rates may have contributed to spatial variation in gestational age, but it seems unlikely that they are an important driver of this variation.

While young and old maternal age are well-known predictors of shorter gestation,[30 31] the association we found with missing data on the father's nationality was unexpected. In the vast majority of cases, the information is missing because no father came forward and officially accepted paternity of the child. It is possible that missing data about the father are an indicator of lower socioeconomic position and social support of the mother, resulting in greater vulnerability. Studies from the USA found a missing name of the father on the infant's birth certificate was associated with lower education, smoking during pregnancy, preterm birth, lower birth weight, no breastfeeding and higher neonatal and postneonatal mortality.[32–35] Children not recognised by their fathers may thus be a group at higher risk of infant and child morbidity and mothers might benefit from additional care during pregnancy and postnatally.

There are several limitations to our study. We did not have data about maternal health-related behaviours such as smoking,[36] mothers' weight and height,[36] disease such as gestational diabetes and data on parental genetic factors. While parental nationality and education might have served as crude proxies for some missing variables, individual-level data about these factors would be valuable. A recent large-scale meta-analysis of genome-wide association data indicated that genetic factors influence birth weight through their effects on gestational age, maternal glucose metabolism, cytochrome P450 activity and possibly through effects on maternal immune function and blood pressure.[37] Of note, compared with the fetus who carries maternal and paternal genes, maternal genes exert a larger effect on gestational age and a weaker effect on birth weight.[38 39]

Our study also showed associations between mean gestational age and the proportion of preterm births (<37 weeks), as well as mean birth weight and proportion of low birth weight newborns (<2500 g) across the 705 small areas, that is, associations with conditions that are clinically relevant (online supplementary figure S3). However, from a statistical point of view, analysing means is more robust and powerful than using a binary indicator defined by a cut-off.[40] Finally, we adjusted analyses of birth weight for gestational age, which may mediate the effects of other variables, for example, maternal age. Adjusting for a variable on the causal pathway has been criticised because it may introduce selection bias (or collider bias in the language of directed acyclic graphs), if there are unknown or unmeasured factors that have an effect on both gestational age and birth weight.[41–43] In our study, results were broadly similar with and without adjustment

for gestational age and the focus of our study was not on causal inference, but on gaining an understanding of the factors contributing to spatial variation of birth weight and gestational age.

In conclusion, our study identified important differences in mean gestational age and birth weight across Switzerland. Small area variation in birth weight is largely, and gestational age partially, explained by pregnancy-related, parental and environmental factors.

**Contributors** ME and CEK conceived the study and obtained funding. VS, ME, MZ and CEK developed the analysis plan. VS did all statistical analyses and wrote the first draft of the paper, which was revised by ME taking into account the critical comments from CEK, MZ, MA and NL. ME supervised the study. All authors approved the final version of the report.

**Funding** The Swiss National Cohort is funded by the Swiss National Science Foundation (SNSF) cohort grant No. 148415. The current analysis was funded by SNSF project grant No. 163452. ME was supported by special SNSF project funding (grant No. 174281).

**Map disclaimer** The depiction of boundaries on this map does not imply the expression of any opinion whatsoever on the part of BMJ (or any member of its group) concerning the legal status of any country, territory, jurisdiction or area or of its authorities. This map is provided without any warranty of any kind, either express or implied.

**Competing interests** None declared.

**Patient consent for publication** Not required.

**Ethics approval** The SNC has been approved by the Ethics Committee of the Canton of Bern.

**Provenance and peer review** Not commissioned; externally peer reviewed.

**Data availability statement** Data are available upon reasonable request. Data may be obtained from a third party and are not publicly available.

**ORCID iDs**
Marcel Zwahlen http://orcid.org/0000-0002-6772-6346
Nicola Low http://orcid.org/0000-0003-4817-8986
Matthias Egger http://orcid.org/0000-0001-7462-5132

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
