## [Reviewer comments · BMJ Open]

ARTICLE DETAILS

TITLE (PROVISIONAL)	Spatial epidemiology of gestational age and birth weight in Switzerland: Census-based linkage study
AUTHORS	Skrivankova, Veronika; Zwahlen, Marcel; Adams, Mark; Low, Nicola; Kuehni, Claudia; Egger, Matthias

VERSION 1 – REVIEW

REVIEWER	Richard McNally Newcastle University, UK
REVIEW RETURNED	14-Dec-2018

GENERAL COMMENTS	This is a very well written paper.
------------------------------------

REVIEWER	Jorge Martinez INECOA, Universidad Nacional de Jujuy, CONICET. Jujuy, Argentina.
REVIEW RETURNED	18-Dec-2018

GENERAL COMMENTS	Comments to the Editor This study represents a significant contribution to the explanation of the factors that affect the period of gestation and birth weight. I believe that the statistical tests were correctly used but I suggest the intervention of an expert to make specific contributions. Comments to the authors: The fact of combining the databases and comparing them is very creative. here are not many studies that have these characteristics. The use of statistics in my opinion is impeccable. I would suggest for next studies to reduce the number of variables to present since the tables are, in some cases, very extensive. The supplementary material was very helpful, but it would add to the maps the neighboring countries and the northern direction to better locate the reader.
---

REVIEWER	Jennifer Zeitlin Inserm, Paris
REVIEW RETURNED	19-Jan-2019

GENERAL COMMENTS	This paper looks at the spatial variation of gestational age and birthweight in Switzerland and seeks to analyze how difference in sociodemographic and geographic characteristics explain the observed variations. One of the difficulties, acknowledged by the authors, is the absence of some key information as well as having variables defined for demographic purposes which only imperfectly capture perinatal risk (no data on parity, but rather firstborn in the
---

	marriage, for instance). Limitations in the data also led to restricting some of the analysis to a sub-sample of births to married women. The paper is well written, has a comprehensive discussion detailing potential limitations and justifying the methodological choices. However, I struggled with this paper because, although the authors make a compelling case for exploring spatial differences in health outcomes in their introduction, the exact hypotheses that they were seeking to address were not sufficiently explained. In particular, the authors did not link the spatial differences observed in their country to characteristics of the environment (economic or social), which might have explained why the differences in language region were the principal factor related to the variation. Would it have been possible to add small area data from other sources on income, unemployment, smoking prevalence, etc, that might have captured the differences between the regions? The principal individual-level factors contributing to explaining variation in gestational age and birthweight are well-known and observed in most studies. Given these known associations, it was not clear whether the authors found that the proportion of variation explained was higher or lower than expected. Furthermore, the use of the outcome gestational age (GA) in days is unusual, as the relationship with health outcomes (such as mortality or neurodevelopment) is primarily noted for births occurring below 37 weeks, extending for some outcomes to under 39 weeks. This outcome is rarely investigated linearly in days (and there is likely not a linear association with health between 39 and 41 weeks and, in fact, risk rises again at 42 weeks; similarly with birthweight, risks rise at the higher end of the birthweight spectrum as well and there is a high amount of normal variation in average birthweight). The complexity of these relationships can be observed for the impact of primiparity (even though imperfectly measured, as described above) on the two outcomes, as it lengthens gestation (primiparity is associated with both earlier and later deliveries, but the latter impact seems to be predominant, on average), but it decreases birthweight. In supplementary material, the authors do assess the correlations between mean birthweight and % preterm and % low birthweight (although one could argue that this should be % small for gestational age), but these are clearly – although correlated – distinct conceptual outcomes. Because medical practices related to induction and caesarean section are likely related to these differences, it is too bad that this information was not available. The authors claim that “Whilst Caesarean section rates vary geographically, they are unlikely to account for the observed spatial variation in gestational age at birth. Geographical patterns of Caesarean section are largely driven by urban-rural differences.” The reference for this statement is a map of caesarean rates in Switzerland, which is hard to analyse in relation to the results presented. Given the large impact that obstetrical practices (caesarean, but also induction) can have on average gestational age (which will have a corresponding effect on birthweight), this potential hypothesis requires more consideration.
--	---

VERSION 1 – AUTHOR RESPONSE

Reviewer: 1

Reviewer Name: Richard McNally

Institution and Country: Newcastle University, UK Please state any competing interests or state 'None declared': None declared.

Please leave your comments for the authors below:

This is a very well written paper.

Authors' response: Thank you.

Reviewer: 2

Reviewer Name: Jorge Martinez

Institution and Country: INECO, Universidad Nacional de Jujuy, CONICET. Jujuy, Argentina.

Please state any competing interests or state 'None declared': None declared

Please leave your comments for the authors below:

The fact of combining the databases and comparing them is very creative. Here are not many studies that have these characteristics. The use of statistics in my opinion is impeccable. I would suggest for next studies to reduce the number of variables to present since the tables are, in some cases, very extensive. The supplementary material was very helpful, but it would add to the maps the neighboring countries and the northern direction to better locate the reader.

Authors' response: Thank you for the comments. The orientation of the map is standard, with North up. We added a note to the figure description to clarify this. We believe that adding the maps of the neighbouring countries would make the graphs unnecessarily busy.

Reviewer: 3

Reviewer Name: Jennifer Zeitlin

Institution and Country: Inserm, Paris

Please state any competing interests or state 'None declared': None declared

Please leave your comments for the authors below:

This paper looks at the spatial variation of gestational age and birthweight in Switzerland and seeks to analyze how difference in sociodemographic and geographic characteristics explain the observed variations.

1. Parity

One of the difficulties, acknowledged by the authors, is the absence of some key information as well as having variables defined for demographic purposes which only imperfectly capture perinatal risk (no data on parity, but rather firstborn in the marriage, for instance). Limitations in the data also led to restricting some of the analysis to a sub-sample of births to married women.

Authors' response: Thank you for your comments and detailed suggestions for improvement of our manuscript. We agree that the variable "parity rank within current marriage" is a poor substitute for the actual biological parity rank. We are thus happy to report that this problem was resolved. We were not aware that the data on birth rank are in fact now available. We obtained them from the Federal Office of Statistics and have rerun all analyses. The complete case population now includes also non-married mothers, and models 3 & 4 for both populations now include civil status. Of note, the coefficient estimates for the other predictors are similar to the previous version.

We made numerous changes (marked in yellow) in the paper that reflect the new analyses.

2. Small-area socio-economic factors

The paper is well written, has a comprehensive discussion detailing potential limitations and justifying the methodological choices. However, I struggled with this paper because, although the authors make a compelling case for exploring spatial differences in health outcomes in their introduction, the exact

hypotheses that they were seeking to address were not sufficiently explained. In particular, the authors did not link the spatial differences observed in their country to characteristics of the environment (economic or social), which might have explained why the differences in language region were the principal factor related to the variation. Would it have been possible to add small area data from other sources on income, unemployment, smoking prevalence, etc, that might have captured the differences between the regions? The principal individual-level factors contributing to explaining variation in gestational age and birthweight are well-known and observed in most studies. Given these known associations, it was not clear whether the authors found that the proportion of variation explained was higher or lower than expected.

Authors' response: We believe a strength of this paper is the detailed examination of the percentage of spatial variation explained by each variable, as shown in Table 3 and Figure 2. We agree with the referee that exploring the influence of neighborhood characteristics is of interest in this context, and we have now done this by including the Swiss neighbourhood index of socio-economic position (SEP) [1]. This index was calculated for each residential building, based on a neighborhood of the 50 closest households, combining information on rent per square meter, number of persons per room, education and occupation of heads of households from the 2000 census. It has recently been updated using data from the Swiss Structural Surveys 2010-2014. The updated index combines information on number of persons per household, mean yearly income, ability to save money and self-assessment of the financial situation. We found that the index of neighbourhood SEP contributed little to explaining the spatial variation in gestational age and birth weight (0% to 2%). In order not to extend the tables further (see comment by referee 2), we decided to present these results in the context of a sensitivity analysis in a supplementary table (see new supplementary Table S4). We acknowledge that the index does not include information on smoking, but we have previously shown that it correlates with mortality from causes associated with socioeconomically patterned behaviours, such as smoking or diet [1].

Again, the changes made in the paper are marked in yellow.

3. Continuous vs. dichotomous Gestational Age

Furthermore, the use of the outcome gestational age (GA) in days is unusual, as the relationship with health outcomes (such as mortality or neurodevelopment) is primarily noted for births occurring below 37 weeks, extending for some outcomes to under 39 weeks. This outcome is rarely investigated linearly in days (and there is likely not a linear association with health between 39 and 41 weeks and, in fact, risk rises again at 42 weeks; similarly with birthweight, risks rise at the higher end of the birthweight spectrum as well and there is a high amount of normal variation in average birthweight). The complexity of these relationships can be observed for the impact of primiparity (even though imperfectly measured, as described above) on the two outcomes, as it lengthens gestation (primiparity is associated with both earlier and later deliveries, but the latter impact seems to be predominant, on average), but it decreases birthweight. In supplementary material, the authors do assess the correlations between mean birthweight and % preterm and % low birthweight (although one could argue that this should be % small for gestational age), but these are clearly – although correlated – distinct conceptual outcomes.

Authors' response: We believe there is a misunderstanding here regarding the scope and aims of this our study. As we say at the end of the Introduction “The objectives of this study were to conduct a nationwide analysis of spatial variation in gestational age and birth weight, and to assess how much small-area variation was explained by available data about neonatal and pregnancy-related variables, parental characteristics and geographical variables”.

We acknowledge that changes in proportion of preterm births might be clinically more relevant than the changes in mean gestational age, especially in case of U-shaped relationships between predictors and the outcome. We agree with the referee that there is “likely not a linear association with health” but please note that we did not consider the association with health status in our study. We stress that this is an epidemiological study, not a clinical one, and our aim was not to inform clinical practice, but to analyze the extent and explanation of spatial variation of gestational age and birth weight.

For this purpose, the analysis of mean gestational age is more appropriate than the use of a binary indicator defined by a (somewhat arbitrary) cutoff of preterm delivery. Altman and Royston

discuss in detail why dichotomizing a continuous variable is usually not a good idea [2]. Firstly, dichotomizing a continuous variable reduces information and the power to detect differences. Newborns close to but on opposite sides of the threshold are characterized as being very different rather than very similar. On the other hand, newborns on the same side of the threshold are characterized as being the same while they might be in fact very different, i.e., not all preterm babies are equally close to being “term”. Comparing the odds of preterm births only captures differences across the threshold and does not reflect their extent across the range gestational age. Finally, please note that we carefully considered the shape of associations and modelled some variables using splines (birth weight) or piece-wise linear functions (maternal age). We did not simply assume linear associations.

4. C-section

Because medical practices related to induction and caesarean section are likely related to these differences, it is too bad that this information was not available. The authors claim that “Whilst Caesarean section rates vary geographically, they are unlikely to account for the observed spatial variation in gestational age at birth. Geographical patterns of Caesarean section are largely driven by urban-rural differences.” The reference for this statement is a map of caesarean rates in Switzerland, which is hard to analyse in relation to the results presented. Given the large impact that obstetrical practices (caesarean, but also induction) can have on average gestational age (which will have a corresponding effect on birthweight), this potential hypothesis requires more consideration.

Authors’ response: We agree that the lack detailed data (on the individual or small area level) is an important limitation of our study, and we acknowledge this in the Discussion. However, as we explain in the Discussion, we would expect regions with higher proportions of Caesarian section to have lower mean gestational age (and consequently birthweight). The geographical data available on Caesarian section do not show this [3]: highly populated, urban areas such as Zurich, Bern or Basel have some of the highest rates of Caesarian sections (probably many pre-planned) while also having relatively high mean gestational age and birth weight. On the other hand, some rural areas, for example in the Canton of Neuchâtel, have low section rates and also low mean gestational ages. We now discuss this in greater detail as follows:

“In particular, data at the individual or small area level on the mode of delivery (vaginal or by Caesarean section, induced or spontaneous) were not available. The proportion of live births with Caesarian section as the mode of delivery varies across regions in Switzerland, and it is reasonable to expect that it would explain some of the remaining variation, both in gestational age and birth weight. Specifically, we would expect regions with higher proportions of Caesarian section to have lower mean gestational age (and consequently birthweight). However, the regional rates of Caesarian section published by the Federal Office of Statistics do not match this expectation [29], with urban areas showing some of the highest Caesarian section rates but also high mean gestational age and birth weight. In fact, geographical patterns of Caesarean section seem to be largely driven by urban-rural differences. Differences in section rates may have contributed to spatial variation in gestational age, but it seems unlikely that they are an important driver of this variation.”

References:

- 1 Panczak R, Galobardes B, Voorpostel M, *et al.* A Swiss neighbourhood index of socioeconomic position: development and association with mortality. *J Epidemiol Community Health* 2012;**66**:1129–36. doi:10.1136/jech-2011-200699
- 2 Altman DG, Royston P. The cost of dichotomising continuous variables. *BMJ* 2006;**332**:1080.
- 3 Federal Office of Statistics. Rate of Caesarian section (MS regions). <https://www.bfs.admin.ch/bfs/en/home/statistics/catalogues-databases/maps.assetdetail.4262550.html>

VERSION 2 – REVIEW

REVIEWER	Richard McNally Newcastle University, UK
REVIEW RETURNED	21-Jun-2019
GENERAL COMMENTS	This paper is now suitable for publication.